

# Genetic diversity of *Poa pratensis* L. depending on geographical origin and compared with genetic markers

Magdalena Szenejko[1,2], Przemysław Śmietana[1,2] and Edyta Stępień[3]

[1] Faculty of Biology, Department of Ecology and Environmental Protection, Institute for Research on Biodiversity, University of Szczecin, Szczecin, Poland
[2] Faculty of Biology, Molecular Biology and Biotechnology Center, University of Szczecin, Szczecin, Poland
[3] Department of Plant Taxonomy and Phytogeography, Institute for Research on Biodiversity, Faculty of Biology, University of Szczecin, Szczecin, Poland

## ABSTRACT

**Background**. *Poa pratensis* is one of the most common species of meadow grass in Europe. Most cultivars of the species found in Poland were originally derived from its ecotypes. We compared the effectiveness of the RAPD and ISSR methods in assessing the genetic diversity of the selected populations of *P. pratensis*. We examined whether these methods could be useful for detecting a possible link between the geographical origin of a given population and its assessed genetic variation.

**Methods**. The molecular markers RAPD and ISSR were used and their efficiency compared using, inter alia, statistical multivariate methods (UPGMA and PCA).

**Results**. The low value of Dice's coefficient (0.369) along with the significantly high percentage of polymorphic products indicates a substantial degree of genetic diversity among the studied populations. Our results found a correlation between the geographical origin of the studied populations and their genetic variations. For ISSR, which proved to be the more effective method in that respect, we selected primers with the greatest differentiating powers correlating to geographical origin.

**Discussion**. The populations evaluated in this study were characterized by a high genetic diversity. This seems to confirm the hypothesis that ecotypes of *P. pratensis* originating from different regions of Central Europe with different terrain structures and habitat conditions can be a source of great genetic variability.

Corresponding author
Magdalena Szenejko,
magdalena.szenejko@usz.edu.pl

## INTRODUCTION

*Poa pratensis* L. is one of the most common species of meadow grass in Europe. It occurs throughout the country and is highly valued as a grass cultivar that can be used for various purposes (*Mirek & Pięknoś-Mirkowa, 2007*; *Szenejko, 2014*). Due to its high plasticity, high spreading capability and significant expansiveness, *P. pratensis* is considered a cosmopolitan species capable of occupying a wide range of different habitats. Its wide range of distribution and the ease with which the species adapts to very different conditions has resulted in a great diversity of ecotypes thriving in a miscellany of habitats. In the case of *P. pratensis*, the selection of particular ecotypes became the basis for breeding work. In Poland, most

cultivars of the species were derived from natural grasslands, often from individual plants. Because apomixis is the dominant manner of reproduction in *P. pratensis,* achieving high genetic variability within individual populations is rather difficult (*Müntzing, 1933*; *Muller, 1964*; *Felsenstein, 1974*; *Pamilo, Nei & Li, 1987*; *Huff & Bara, 1993*). In addition to vegetative and sexual reproduction processes, the plant can also reproduce asexually from seeds, which are formed through apospory or diploid parthenogenesis and without any fertilization process (*Mazzucato, Den Nijs & Falcinelli, 1996*; *Albertini et al., 2001*; *Spillane, Steimer & Grossniklaus, 2001*; *Albertini et al., 2005*; *Matzk et al., 2005*). As a result, offspring and parent plants may be genetically identical (*Albertini et al., 2004*; *Carneiro, Dusi & Ortiz, 2006*). This implies that the initial forms i.e., the different ecotypes and cultivars used for breeding did not produce new combinations of genes. As a result, cultivars derived from such initial material have similar or related genotypes. The results of our earlier research, which concerned the evaluation of genetic variability in cultivars and ecotypes of *P. pratensis* native to Central Europe using RAPD markers, confirmed the considerable genetic relatedness of those cultivars and ecotypes (*Szenejko, Filip & Słominska-Walkowiak, 2009*; *Szenejko & Rogalski, 2015*). The populations evaluated in those studies originated from Lower Silesia and Podlasie, regions of Poland with different habitats and climatic conditions. The ecotypes analyzed in those previous studies differed with respect to their phenotypic traits but showed little genetic variability, whereas the populations in the present study were selected from habitats as diverse as possible, i.e., lowlands, uplands and mountains. The ecotypes in this study originated from the western and southern regions of Poland as well as from the Kujawy and Podlasie regions. It was assumed that long distances between regions would affect habitat conditions, and that natural spatial barriers would cause the greatest genotypic diversity among the research material. In addition, three cultivars (including two of the oldest Polish cultivars of *P. pratensis*) were included in our research and subjected to the same analyses. All the plant material was evaluated for genetic diversity using two systems of markers based on DNA amplification (RAPD and ISSR). In the present study, the effectiveness of the methods in assessing the genetic diversity of the selected forms of *P. pratensis* was compared. Furthermore, we investigated whether these methods could be useful in establishing a link between the geographical origin of a given population and their assessed genetic variability. Primers with the greatest differentiating powers correlating with geographical distance were selected for ISSR, the more effective method in that respect. Principal Component Analysis (PCA) was used for this procedure, which was performed on the chosen values of DNA amplification products obtained in the presence of those selected ISSR markers with the highest genetic differentiating power with respect to the studied forms of *P. pratensis*.

## MATERIALS & METHODS

### Plant material

This assessment of the genetic diversity of 18 selected forms of *P. pratensis* was carried out in 2014. The study involved 3 cultivars and 15 ecotypes selected from locations as distant as possible from each other. The seeds used in the study came from the national collection of the Botanical Garden of the Institute of Plant Breeding and Acclimatization in Bydgoszcz

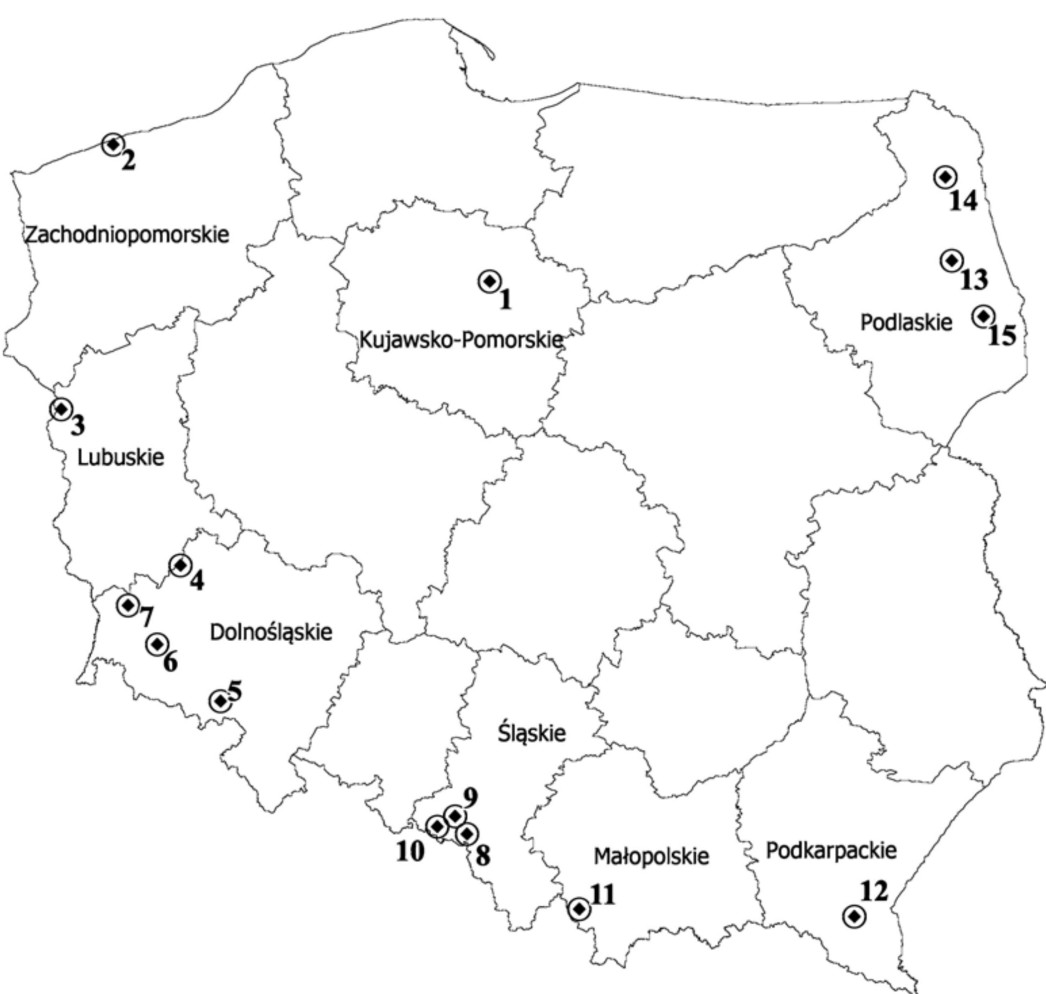

**Figure 1** **Map depicting the locations of the 15 populations of *P. pratensis* used in the study.** 1, pom06 175; 2, zap00 240; 3, lbs05 170; 4, lbs05 628; 5, dos01 333; 6, dos01 405; 7, dos01 468; 8, bes03 373; 9, bes03 379; 10, bes03 35; 11, ora03 178; 12, bie04 189; 13, pod02 215; 14, pod02 172; 15, pod02 317.

(IHAR); the caryopses used for establishing this collection were collected during field trips organized by the staff of the Botanical Garden and originated from different geographical regions of Poland, including 8 different voivodships (provinces) (Fig. 1 and Table 1). The main criteria when harvesting the caryopses were the diversity of plant material and variety of habitats. The seed material was collected from wastelands and arable land (meadows and pastures), as well as from environmentally valuable areas. The assessment of genetic diversity was performed for three cultivars used for different purposes, including 'Eska 46,' one of the oldest Polish fodder cultivar and two lawn cultivars: 'Limousine', an apomictic uniclonal German cultivar, and 'Alicia,' the oldest Polish lawn cultivar.

## DNA isolation

Genomic DNA was isolated from the plant tissue using magnetic beads (Novabeads Plant DNA STANDARD Purification KIT), following the procedure for monocotyledons and the manufacturer's instructions (Novazym, Warszawa, Poland). For each form of *P. pratensis*

**Table 1 Ecotypes of *P. pratensis* analyzed in the study and the locations where their caryopses were collected.**

| Ecotype | Town/Place | Latitude | Longitude | Habitat |
|---|---|---|---|---|
| pom06 175 | Wielkie Stwolno | 53°26′34.1″N | 18°39′25.8″E | Vistula River oxbow lake |
| zap00 240 | Łukęcin | 54°02′58.4″N | 14°52′03.7″E | Beach |
| lbs05 170 | Owczary | 52°28′16.3″N | 14°39′08.0″E | Ostnicowy nature reserve |
| lbs05 628 | Przemków | 51°32′50.6″N | 15°46′33.4″E | Meadow |
| dos01 333 | Rybnica Leśna II | 50°42′08.5″N | 16°18′00.3″E | Pasture |
| dos01 405 | Dziwiszów–Góry Kaczawskie | 50°57′39.4″N | 15°49′23.6″E | Meadow |
| dos01 468 | Oleszna Podgórska | 51°21′02.3″N | 15°28′12.1″E | Meadow |
| bes03 373 | Ustroń–Wielka Czantoria | 49°40′45.6″N | 18°47′42.7″E | Mountain meadow |
| bes03 379 | Ustroń–Polana Goryczkowa | 49°41′05.6″N | 18°46′58.6″E | Meadow pasture |
| bes03 354 | Jaworzynka | 49°31′20.2″N | 18°51′31.2″E | Meadow |
| ora03 178 | Lipnica Wielka | 49°27′05.3″N | 19°37′40.6″E | Meadow |
| bie04 189 | Płonna | 49°26′11.7″N | 22°07′05.9″E | Wasteland |
| pod02 215 | Lipina | 53°19′59.6″N | 23°25′28.5″E | Meadow/pasture |
| pod02 172 | Walne | 53°58′20.0″N | 23°04′04.6″E | Meadow |
| pod02 317 | Pieńki | 53°03′32.3″N | 23°38′47.2″E | Pasture |

100–150 mg of material was taken from 80 random etiolated seedlings. The DNA was isolated in two repetitions.

## RAPD method

RAPD analysis, based on random amplification of polymorphic DNA, was performed according to the modified method described by *Williams et al. (1990)*. Sixty-nine primers were tested and 7 were chosen to evaluate RAPD polymorphism (Table 2). The amplification reactions were performed in a T100$^{TM}$ Thermal Cycler (Bio-Rad Polska). The thermal profile proposed by *Rajasekar, Fei & Christians (2005)* was used: initial denaturation for 5 min at 95 °C, then 94 °C—1 min, 37 °C—90 s, and 72 °C—1 min in 45 cycles, with a final elongation for 7 min. The reactions were carried out on a total volume of 25 µl containing the following ingredients: 1xbufor for PCR reaction with Mg$^{2+}$ (Novazym, Poland), 1.5 mM MgCl$_2$, 0.2 mM dNTPs, 1.0 µM primer, 1.25 U RedAllegro *Taq* Polymerase (Novazym, Warszawa, Poland) and 125 ng of DNA.

## ISSR method

ISSR analysis was performed following the modified method described by *Ziętkiewicz, Rafalski & Labuda (1994)* and the experimental conditions optimized by *Wang (2010)*. Having pretested 66 microsatellite primers, 13 of these (Oligo IBB PAN, Poland) were chosen to evaluate ISSR polymorphism (Table 2). The reaction mixture, 25 µl in volume, contained: 1xbufor for the PCR reaction with Mg$^{2+}$ (Novazym, Poland), 1.5 mM MgCl$_2$, 0.25 mM dNTP, 0.4 µM of primer 1 U RedAllegro *Taq* Polymerase (Novazym, Poland) and 50 ng of DNA.

The amplification reactions were carried out in a T100$^{TM}$ Thermal Cycler (Bio-Rad Polska) for 40 cycles. The ISSR PCR reaction was as follows: initial denaturation at 94 °C

**Table 2  The sequences of primers used in the study.** Random amplification of polymorphic DNA was held in the presence of 7 10-nucleotide primers manufactured by Oligo IBB PAN, Poland. 13 microsatellite primers were used for the ISSR-PCR reaction. Most of them were dinucleotides composed of 17 and 18 bp.

| Primer RAPD | Sequence 5′–3′ | Primer ISSR | Sequence 5′–3′ |
|---|---|---|---|
| B10 | CTGCTGGGAC | 807 | $(AG)_8T$ |
| B17 | AGGGAACGAG | 808 | $(AG)_8C$ |
| C16 | CACACTCCAG | 810 | $(GA)_8T$ |
| F05 | CCGAATTCCC | 811 | $(GA)_8C$ |
| G19 | GTCAGGGCAA | 834 | $(AG)_8YT$ |
| M14 | AGGGTCGTTC | 840 | $(GA)_8YT$ |
| P08 | ACATCGCCCA | 841 | $(GA)_8YC$ |
| | | 857 | $(AC)_8YG$ |
| | | 888 | $BDB(CA)_7$ |
| | | R07 | $(AC)_8YT$ |
| | | M09 | $(AC)_8YG$ |
| | | N01 | $CA(GT)_8$ |
| | | TGA | $(TGA)_6A$ |

Notes.

**Y**, T or C; **B**, C, G or T; **D**, A, G or T.

for 4 min, then 94 °C—30 s, 52 °C—45 s, 72 °C—2 min, with a final elongation carried out at 72 °C for 7 min.

## Data analysis

The resulting RAPD and ISSR amplification products were separated on 2.0% agarose gel with ethidium bromide (5 μg/ml; Sigma-Aldrich) in a TAE buffer for 4–5 h. Visualization, documentation and analysis of the results were carried out using Gel Doc™ XR+ and Quantity One 4.6.5 software manufactured by Bio-Rad. The Polymorphism Information Content (PIC) index for dominant primer systems was calculated for both RAPD and ISSR (*Ghislain et al., 1999*). The Assay Efficiency Index (AEI) value was also calculated (*Pejic, Ajmone-Marsan & Morgante, 1998*). The Genetic Similarity Index (Si) of the studied ecotypes and cultivars of *P. pratensis* was determined in accordance with Dice's coefficient (*Dice, 1945*) following *Nei & Li (1979)*. The correlations between them were determined on the basis of genetic similarity matrices obtained in the presence of the RAPD and ISSR markers by calculating Spearman's rank correlation coefficient ($r$; $p < 0.05$); then the determined genetic similarity matrix was used to construct UPGMA dendrograms in FreeTree and TreeView 1.6.6 software (*Pavlíček, Hrdá & Flegr, 1999*; *Hampl, Pavlíček & Flegr, 2001*). To analyze the quality of information from particular RAPD and ISSR primers, Principal Component Analysis (PCA) was used. The information on the size of the selected amplification products obtained in the presence of each primer was used as the initial data; then the PCA assessment was carried out by reducing three-dimensional space to two dimensions. The results of multiple analyses were compared to the PCA results. The PCA results obtained using the initial data, in the form of information on the geographical
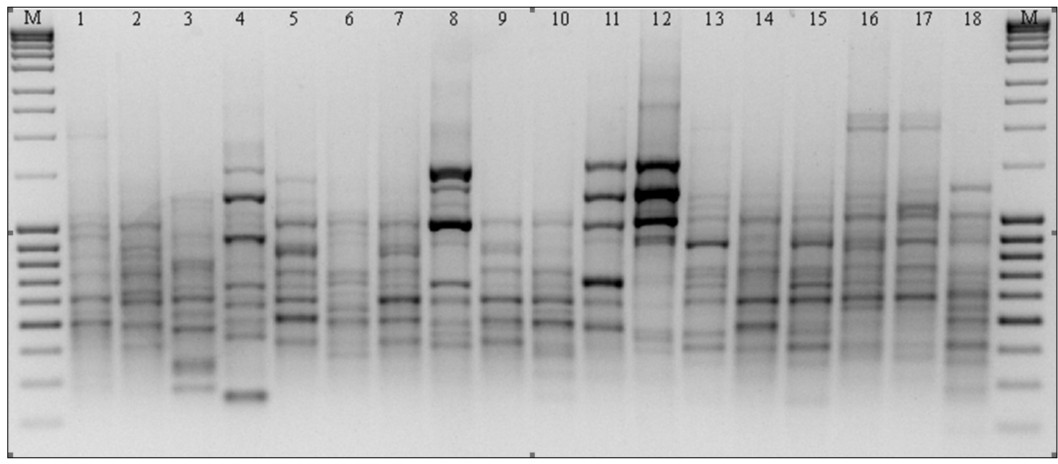

**Figure 2   RAPD patterns amplified with primer M14 for *P. pratensis*.** M, MassRuler™ Ladder mix, ready-to-use (Thermo Scientific); 1, pom06 175; 2, zap00 240; 3, lbs05 170; 4, lbs05 628; 5, dos01 333; 6, dos01 405; 7, dos01 468; 8, bes03 373; 9, bes03 379; 10, bes03 354; 11, ora03 178; 12, bie04 189; 13, pod02 215; 14, pod02 172; 15, pod02 317; 16, 'Alicia'; 17, 'Limousine'; 18, 'Eska 46.'

location of the place of origin of particular ecotypes, was used as the target matrix. These analyses were carried out using the STATISTICA 10.0 PL statistical package.

## RESULTS

### RAPD analysis

In the course of RAPD analysis, in total 246 amplification products were obtained for which 100% polymorphism was demonstrated (Table 3). The primers chosen to evaluate polymorphism generated bands in a wide size range: from 149 bp to 2,367 bp in the presence of primer C16. The greatest number of amplicons (43) was obtained as the result of reactions with primers G19 and M14 (Fig. 2). The primers used for the analysis initiated the synthesis of 79 specific amplification products which differentiated the studied forms of *P. pratensis*. Of these, the largest number of DNA fragments (16) which had a unique composition were obtained in reaction with the M14 primer (Fig. 2). As many as 68 specific products were found for the ecotypes, with the largest number of them in a population from Silesia (bes03 373). In the presence of six primers, we were able to generate 15 products with a unique composition, ranging in size from 149 bp (for C16) to 1478 bp (for M14). Unique DNA fragments of approximately the same size were identified in amplification reactions with different primers. A significant number of specific bands were also identified for two ecotypes, one from Podkarpacie (bie04 189) and the other originating from the borderland region of Lubuskie and Lower Silesian voivodships (provinces) (lbs05 628) 11 and 10 bands respectively.

### ISSR analysis

As the result of amplification reactions involving 13 microsatellite primers, a total of 514 products ranging in size from 303 bp (for 811) up to 3,105 bp (for 808) were obtained (Table 3 and Fig. 3). In reactions with primers N01 and 840 a large number of bands were

**Table 3** RAPD and ISSR primers used, total number of scored bands (NSB), number of polymorphic bands (NPB), their unique bands (NUB) and percentage of polymorphism bands (PPB) for each primer.

| Name of primer | Size range (bp) | NSB | NPB | NUB | PPB |
|---|---|---|---|---|---|
| | | **RAPD** | | | |
| B10 | 213–1,867 | 35 | 35 | 11 | 100 |
| B17 | 209–1,483 | 32 | 32 | 7 | 100 |
| C16 | 149–2,367 | 33 | 33 | 15 | 100 |
| F05 | 316–2,366 | 34 | 34 | 11 | 100 |
| G19 | 201–1,902 | 43 | 43 | 9 | 100 |
| M14 | 226–2,243 | 43 | 43 | 16 | 100 |
| P08 | 179–1,173 | 26 | 26 | 10 | 100 |
| Total | 149–2,367 | 246 | 246 | 79 | – |
| | | **ISSR** | | | |
| 807 | 341–1,974 | 23 | 23 | 1 | 100 |
| 808 | 325–3,105 | 43 | 43 | 17 | 100 |
| 810 | 338–2,523 | 35 | 35 | 8 | 100 |
| 811 | 303–2,748 | 40 | 40 | 10 | 100 |
| 834 | 404–2,686 | 35 | 34 | 7 | 97 |
| 840 | 373–2,967 | 50 | 50 | 8 | 100 |
| 841 | 373–2,504 | 37 | 36 | 7 | 97 |
| 857 | 380–2,577 | 46 | 45 | 7 | 98 |
| 888 | 501–2,414 | 26 | 24 | 5 | 92 |
| R07 | 576–2,327 | 37 | 37 | 13 | 100 |
| M09 | 353–2,096 | 47 | 47 | 9 | 100 |
| N01 | 404–2,256 | 51 | 51 | 15 | 100 |
| TGA | 362–2,209 | 44 | 43 | 17 | 98 |
| Total | 303–3,105 | 514 | 508 | 124 | – |

generated (51 and 50 respectively), whereas in the presence of primer 807 only 23 products (the smallest number) were obtained. A high level of polymorphism was observed in the amplified DNA fragments and in the presence of eight primers only polymorphic products were obtained. We identified a total of 124 specific amplification products in the ecotypes and cultivars of *P. pratensis* analyzed in this study. The largest number (17) was obtained in reaction with primer 808, while in the presence of primer 807 only one product with a unique composition (649 bp) for ecotype lbs05 170 was obtained (Fig. 3). In total, 95 such products were identified for the ecotypes. The largest numbers were recorded for populations lbs05 170 (15 bands) and lbs05 628 (13 bands), and for ecotype pom06 175. Specific products of similar molecular size were identified in the presence of different primers. A significant number of unique bands were also identified for the lawn cultivars, including the German 'Limousine' (11).

## Evaluation of RAPD and ISSR polymorphisms

Using RAPD and ISSR markers, a significant number of specific amplification products were obtained. 98.6% of the bands obtained using ISSR were polymorphic, whereas
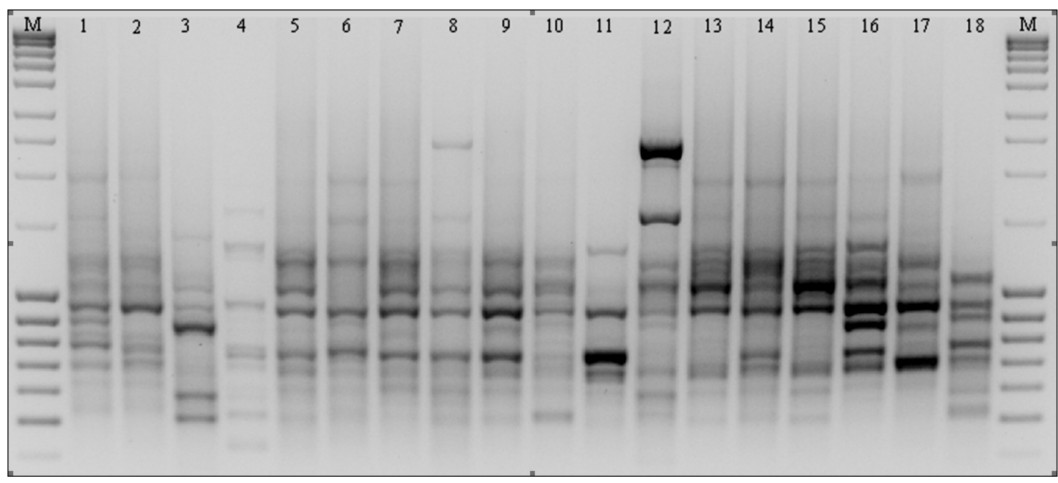

**Figure 3** **ISSR patterns amplified with primer 808 for *P. pratensis*.** M, MassRuler^TM Ladder mix, ready-to-use (Thermo Scientific); 1, pom06 175; 2, zap00 240; 3, lbs05 170; 4, lbs05 628; 5, dos01 333; 6, dos01 405; 7, dos01 468; 8, bes03 373; 9, bes03 379; 10, bes03 354; 11, ora03 178; 12, bie04 189; 13, pod02 215; 14, pod02 172; 15, pod02 317; 16, 'Alicia'; 17, 'Limousine'; 18, 'Eska 46.'

those obtained in RAPD reactions were 100% polymorphic (Table 3). A large number of products were obtained using both methods in the presence of single primers (on average 30 products for each method) (Table 4). For both RAPD and ISSR methods the polymorphism information content (PIC) was calculated; the average PIC value for each method was almost identical and was 0.264 for RAPD and 0.270 for ISSR. The value of the assay efficiency index, which indicates the average number of polymorphic products identified in the presence of a single primer, was also calculated for both RAPD and ISSR marker systems. The obtained AEI value was higher for ISSR markers. Specific DNA fragments were obtained for almost all the studied populations (Tables 3 and 4); RAPD primers were more effective in identifying them. In a reaction with a single RAPD primer, we were able to identify as many as 11 unique bands. Dice's genetic similarity matrix was determined for all the studied forms of *P. pratensis*. Independently of each other, the RAPD and ISSR methods revealed little genetic relationship between the studied populations and cultivars (Table 4). On average Dice's coefficient for RAPD was 0.371, while it was 0.367 for ISSR markers. The greatest genetic relationship was demonstrated between two populations from Podlaskie voivodship, pod02 215 and pod02 317 (0.733 for RAPD, 0.650 for ISSR and 0.678 for RAPD + ISSR). The lowest values of Dice's coefficient for RAPD (0.154) were obtained for populations lbs05 170 and bie04 189, while in the cases of ISSR and RAPD + ISSR the smallest degree of genetic relatedness was found between ecotype lbs05 628 and the cultivar 'Alicia'. The following ecotypes were found to be most genetically distant: bie04 189, lbs05 628, and lbs05 170. To highlight the compatibility of the results of the RAPD and ISSR methods, the correlation between them was calculated. Using Spearman's coefficient (0.461), the statistical correlation between the genetic similarity matrices obtained on the basis of electrophoretic images for RAPD and ISSR was calculated (Fig. 4).
**Table 4  Evaluation of RAPD and ISSR polymorphisms.** The average Number of Scored Bands (NSB) including Number of Unique Bands (NUB) obtained in the presence of a single primer, the value of Polymorphism Information Content (PIC), Assay Efficiency Index (AEI), and Genetic Similarity (Si).

| Parameter/Index | RAPD | ISSR | RAPD + ISSR |
|---|---|---|---|
| NSB | 35.1 | 38.0 | 36.8 |
| NUB | 11.3 | 7.7 | 9.3 |
| AEI | 35.1 | 37.3 | 36.4 |
| PIC | 0.264 | 0.270 | 0.268 |
| Si | 0.371 | 0.367 | 0.369 |
| (Range) | (0.154–0.733) | (0.218–0.650) | (0.222–0.678) |

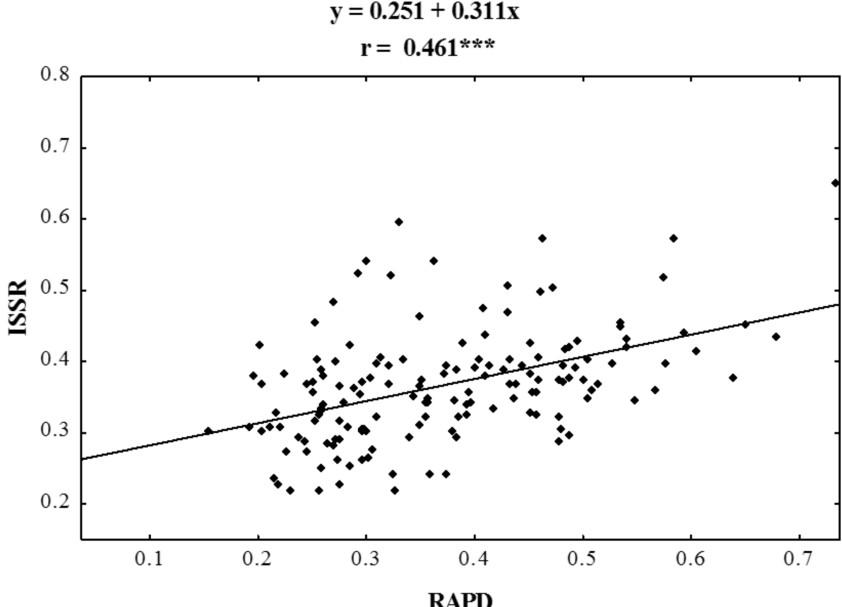

$$y = 0.251 + 0.311x$$
$$r = 0.461^{***}$$

**Figure 4  Correlation analysis of similarity matrices obtained using RAPD and ISSR markers in ecotypes of *P. pratensis*.** The symbol *** indicates that the value is significant assuming $P = 0.001$ as the level of significance.

## Cluster analysis

The applied measure of genetic similarity was used to construct UPGMA dendrograms (Fig. 5). There was greater compatibility with respect to the clustered forms for the ISSR and RAPD + ISSR dendrograms. Regardless of which method was used, two asymmetrical and distinct major clusters were clearly distinguishable in each dendrogram. The majority of the studied forms were assigned to the second cluster, within which two subclusters of similarity were distinguishable 11 forms for RAPD and RAPD + ISSR and 12 forms for ISSR. In the ISSR dendrogram, the first main cluster included two ecotypes from Kujawsko-Pomorskie (pom06 175) and from West Pomerania voivodships (zap00 240). The combined RAPD + ISSR analysis also included ecotype dos01 405 from Lower Silesia. In the case of RAPD, the first main cluster contained two other ecotypes (bie04 189 and ora03 178), while pom06 175 and zap00 240 were assigned to the second main cluster

and to both common subclusters (Fig. 5). Two ecotypes from Lubuskie voivodship (lbs05 170 and lbs05 628) turned out to be genetically most distant. These were not assigned to any pair or subcluster; this is visible in all the dendrograms (Fig. 5). These populations also differed among themselves; this was confirmed by the value of Dice's similarity index (0.224 for RAPD + ISSR). In the ISSR and RAPD + ISSR dendrograms, other genetically different forms included an ecotype from Podkarpacie region (bie04 189), while in the RAPD dendrogram, a form from Silesia (bes03 373) was genetically different. Among the cultivars, 'Eska 46' was the most distinctly different: in the RAPD and ISSR dendrograms, it was connected loosely with the second cluster, while the image obtained for RAPD + ISSR depicted it as an unpaired form. In each of the dendrograms the ecotypes from Podlasie and the lawn cultivars 'Alicia' and 'Limousine' formed a common subcluster of forms. In the RAPD dendrogram, the Silesian population bes03 354 joined them, whereas in the ISSR and RAPD + ISSR dendrograms this form was included in the second subcluster together with other ecotypes from Silesia and populations from Lesser Poland and Lower Silesia. In the ISSR diagram, dos01 405 was placed with ecotypes from Lower Silesia, while in the RAPD diagram it was only loosely connected with the second cluster and in the RAPD + ISSR diagram it was included in a different cluster.

## Principal component analysis

We examined the position of the ecotypes of *P. pratensis* in two-dimensional space using principal component analysis (PCA), which was carried out on the geographical coordinates of the habitats from which the surveyed populations had come. Figure 6A depicts this analysis, which was used as a reference template for the selection of primers with the highest differentiating power. PCA was performed independently for RAPD and ISSR, using the amplification products generated in the presence of individual primers. For each of the methods the four starters were chosen which most clearly and coherently revealed the relationship between the geographic location of the assessed ecotypes and their genetic variability. Subsequently all possible combinations of the selected primers were analyzed for their main components; this was done separately for RAPD, ISSR and (RAPD + ISSR). A total of 60 PCAs were performed for the RAPD and ISSR primers, both for their combinations within one method and also for both methods combined. For the final PCA, DNA amplification products were chosen which had been obtained from the studied ecotypes and cultivars of *P. pratensis* in the presence of ISSR microsatellites 807, 808, 834 and 840; these were the dinucleotide repeat sequences $(AG)_n$ $(GA)_n$, anchored at the end of 3′. The studied populations were distributed on both sides of the axis of coordinates and distinctly away from its center. They formed five separate groups, which indicates their considerable diversity. Three populations from Podlasie (13, 14, 15), and ecotypes from West Pomerania (2) and Kujawy regions (1) together with the lawn cultivar 'Limousine' (17), formed two clearly distinguishable groups (Fig. 6). These were distributed on the same side of the first axis (horizontal) and clearly differed from the remaining forms of *P. pratensis*, which originated from regions in southern and western Poland. Another group, which was located on the opposite side of the second axis, contained two ecotypes (lbs05 628 and dos01 468 (4, 7)) which came from the borderlands of the Lower Silesian and

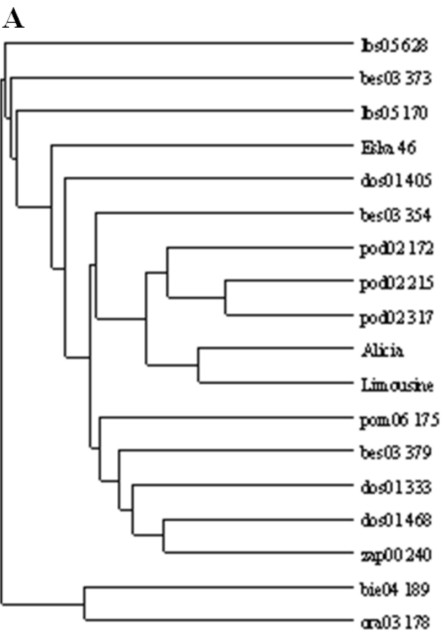

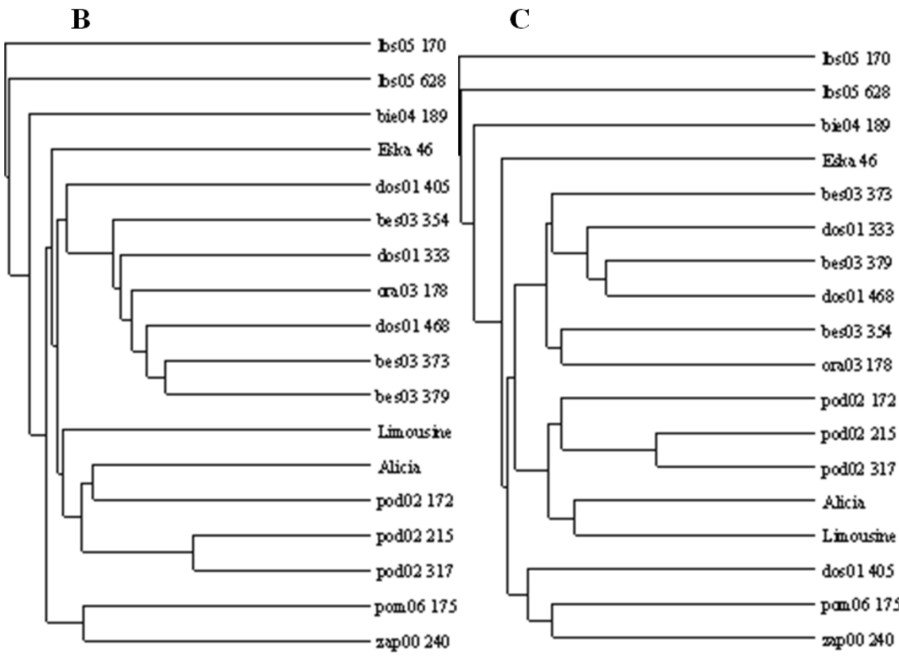

**Figure 5  UPGMA dendrograms of genetic similarity of the studied ecotypes and cultivars of *P. pratensis* obtained on the basis of polymorphism markers constructed using the TreeView 1.6.6.** A, RAPD; B, ISSR; C, (RAPD + ISRR).

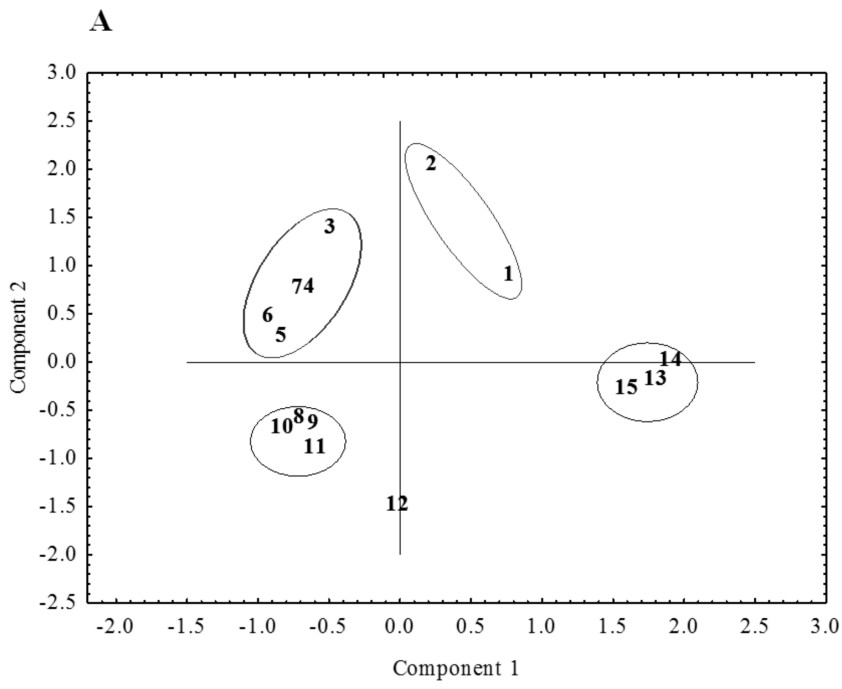

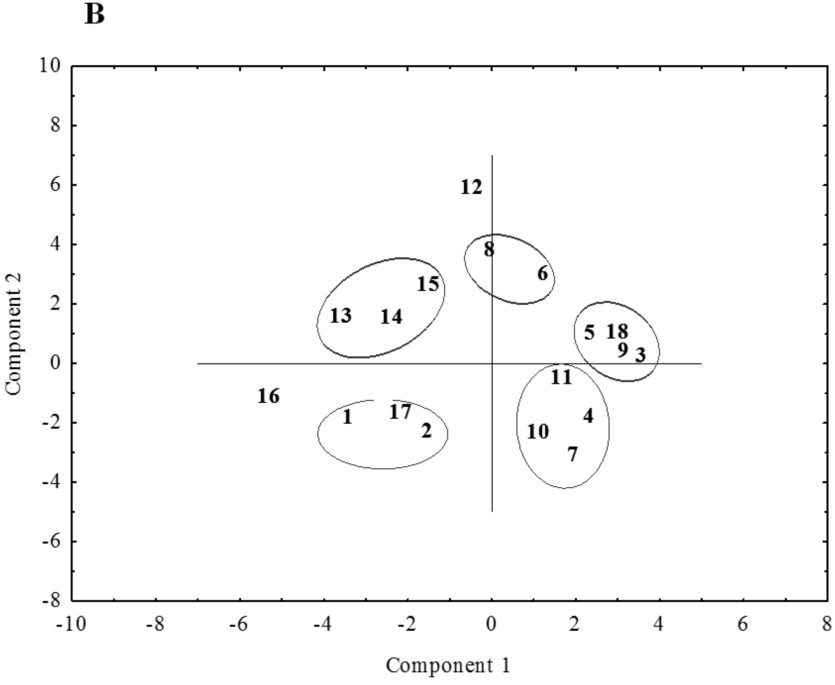

**Figure 6  The distribution of the studied ecotypes and cultivars of *P. pratensis* in two-dimensional space (PCA).** A, the result for the initial data in the form of geographic coordinates of their habitats. B, the final result of PCA analysis using the size of the amplification products generated in the presence of four ISSR primers: 807, 808, 834, 840, the closest to the geographical location of the selected ecotypes: 1, pom06 175; 2, zap00 240; 3, lbs05 170; 4, lbs05 628; 5, dos01 333; 6, dos01 405; 7, dos01 468; 8, bes03 373; 9, bes03 379; 10, bes03 354; 11, ora03 178; 12, bie04 189; 13, pod02 215; 14, pod02 172; 15, pod02 317; 16, 'Alicia'; 17, 'Limousine'; 18, 'Eska 46.'

Lubuskie voivodships respectively. Populations from Silesia (10) and Lesser Poland (11) were also contained in this group. On the same side of the horizontal axis we find another group containing the fodder cultivar 'Eska 46' (18) and three ecotypes (3, 5, 9) from the Lubuskie, Lower Silesian and Silesian voivodships respectively. In the course of the principal component analysis, two forms of *P. pratensis* were identified (12 and 16 ('Alicia')), which were not included in any of the groups. As with the UPGMA dendrograms, the population from Podkarpacie (bie04 189) differed distinctly from the other ecotypes and constituted a distinct genotype (Figs. 5 and 6).

## DISCUSSION

DNA markers have become a useful tool for breeding work and for evaluating the genetic diversity of many different plant species, including the family *Poaceae*. Molecular analysis, in conjunction with the characteristics of morphological traits, may be used to classify plants and select the initial forms for creating new cultivars or breeding lines. In the case of *P. pratensis*, this was confirmed by *Shortell, Meyer & Bonos (2009)*, *Tamkoc & Arslan (2010)*, *Raggi et al. (2015)* and *Yuan et al. (2015)*. In this study, two markers (RAPD and ISSR) based on DNA amplification reactions were used to assess the genetic diversity of ecotypes and cultivars of *P. pratensis*. These methods do not require any prior information on the analyzed DNA sequence. They do not take a lot of time and are also relatively simple and therefore quite universal. Furthermore they require only a small amount of DNA; this is why they are often employed in the genetic evaluation of different plant species. They can be used either in combination (*Nkongolo, Michael & Demers, 2005*; *Alam, Naik & Mishra, 2009*; *De Lima et al., 2011*; *Singh et al., 2011*; *Debajit et al., 2015*) or independently of each other (*Al-Humaid & Motawei, 2004*; *Wang, 2010*; *Guo et al., 2009*; *Bednarskaya et al., 2014*). In grasses, RAPD and ISSR markers are most commonly used to evaluate genetic variability within and between populations, and for molecular characterization and identification of different species and their hybrids, cultivars and genotypes (*Posselt et al., 2006*; *Pivorienė et al., 2008*; *Al-Humaid, Ibrahim & Motawei, 2011*; *Motawei & Al-Ghumaiz, 2012*; *Madesis et al., 2014*; *Yuan et al., 2014*). In *P. pratensis*, genetic analyses employing RAPD and ISSR focus mainly on the identification of sexual and apomictic genotypes (*Huff & Bara, 1993*; *Mazzucato et al., 1995*; *Barcaccia et al., 1997*; *Barcaccia, Veronesi & Falcinelli, 1998*; *Stephens et al., 2006*); the assessment of genetic variability and identification of genotypes and cultivars (*Lickfeldt, Voigt & Hamblin, 2002*; *Ning et al., 2005*; *Liang et al., 2009*; *Fard et al., 2012*; *Wang et al., 2012*; *Yuan et al., 2015*); determining the genetic relationships between different genotypes and species belonging to the genus *Poa* and their hybrids (*Johnson et al., 2002*; *Curley & Jung, 2004*; *Patterson, Larson & Johnson, 2005*; *Goldman, 2008*; *Goldman, 2013*); and identification of pathogens responsible for fungal diseases in grasses (*Hsiang et al., 2000*). Both marker systems can generate high resolution band patterns and high levels of polymorphism as high as 90%. The presence of these markers resulted in a significant number of amplification products in our study: 246 for RAPD and 514 for ISSR. However, ISSR gives better reproducibility of results, which can be explained by its use of longer primers (microsatellites) and higher annealing temperatures

(*Gostimsky, Kokaiva & Konovalov, 2005*; *Reddy, Sarla & Siddiq, 2002*; *Fernández, Figueiras & Benito, 2002*; *Vaillancourt et al., 2008*). Similarly, in previous studies concerning the assessment of the morphological and genetic variability of domestic ecotypes of *P. pratensis* originating from the areas of Lower Silesia and Podlasie (*Szenejko & Rogalski, 2015*), RAPD primers F05, G19 and M14 proved to be most useful in the evaluation of polymorphisms. The highest number of DNA fragments were obtained in the presence of these primers. In the case of ISSR markers, as in the studies of *Bednarskaya et al. (2014)*, better results were obtained in the presence of primers with dinucleotide repeat sequences $(AG)_2,(GA)_2$ and $(AC)_2$, anchored at the end of 3'. As suggested by *Al-Humaid et al. (2004)* and *Al-Humaid, Ibrahim & Motawei (2011)*, dinucleotide and trinucleotide repeats are abundant in genomes of fodder grasses and thus might constitute potential markers for assessing their genetic diversity and identifying their fungal diseases using the ISSR method. In grasses, $(CAC)_n$, $(CTC)_n$ and $(GA)_n$ and $(CA)_n$ are particularly abundant. In the genus *Poa,* ISSR primers with dinucleotide $(AC)_n$ $(CA)_n$ $(GA)_n$ and $(AG)_n$ repeats turned out to be particularly useful for the identification and evaluation of genetic relationships between various species, including *P. angustifolia*, *P. trivialis*, *P. arachnifera*, *P. pratensis*, *P. compressa*, *P. ligularis*, and *P. secunda* and their hybrids (*Goldman, 2008*; *Arslan & Tamkoç, 2011*). A high level of polymorphism was observed in numerous amplification products generated in the course of our analyses. The nature of ISSR, which targets regions especially rich in microsatellites, may also explain the high level of polymorphism, since those regions are known to accumulate a large number of mutations due to DNA polymerase slippage during replication and unequal crossing-over (*Kashi & King, 2006*; *Da Costa et al., 2011*). In the case of ISSR markers, this can lead to longer dinucleotide repeat sequences and insertion into at least one of them. In this way new variants of microsatellite sequences of different lengths appear, which enriches the collection of alleles identified as polymorphic in a population. However, according to many researchers, in the course of RAPD and ISSR analyses significant levels of polymorphism can be generated, and for various forms and species belonging to the genus *Poa* this can exceed 90% (*Liang et al., 2009*; *Arslan & Tamkoç, 2011*; *Fard et al., 2012*; *Wang et al., 2012*; *Cichorz, Gośka & Litwiniec, 2014*). In our analyses, a slightly greater percentage of polymorphic DNA fragments was obtained in reaction with RAPD primers. The data available in the literature indicates that the RAPD method is somewhat superior due to its higher levels of generated polymorphism in the presence of various molecular marker systems which is explained, inter alia, by their polyallelic nature (*Alam, Naik & Mishra, 2009*; *Bhattacharya, Bandopadhyay & Ghosh, 2010*; *Naik et al., 2010*). A possible explanation for the difference in resolution of RAPD and ISSR is that the two-marker techniques target different portions of the genome (*Debajit et al., 2015*). Although a greater percentage of polymorphic DNA products was identified in the presence of RAPD primers, it was microsatellites that we found to be more effective in detecting them. This finding was confirmed by calculating the average efficiency index (AEI), which was slightly higher for ISSR markers. The slight differences in the assessment of RAPD and ISSR polymorphisms did not rule out the possibility of establishing a correlation between them. Even though Spearman's coefficient was not very high (0.461), the result was statistically very significant. Hence, it can be concluded that the results

presented in this study obtained using RAPD and ISSR markers are comparable. *Budak et al. (2004)*, in a study examining the phylogenetic relationships between seed and vegetative biotypes of *Buchloe dactyloides*, used various systems of molecular markers (ISSR, SSR, RAPD and SRAP), for which they determined correlations. In the case of RAPD and ISSR, the Spearman's coefficient (0.461) calculated by Budak was similar to ours, but their results turned out to be statistically insignificant. Both RAPD and ISSR markers revealed a small genetic similarity among the studied ecotypes and cultivars of *P. pratensis*. The average value of Dice's coefficient for both methods combined (RAPD + ISSR) was low (0.369), which together with the high number of polymorphic products may indicate substantial genetic differences between the studied populations and their low level of relatedness. It can be speculated that natural spatial barriers might have inhibited and minimized the transfer of genes among these populations; this would certainly have contributed to their diversity. At the same time, we can surmise that there were populations among the studied ecotypes in which apomixis was the main mode of reproduction. The ecotypes originating from three voivodships (Podkarpackie, Lower Silesian and Lubuskie (bie04 189, lbs05 628 and lbs05 170)), were determined to be those genetically most distant from the other forms. These populations had the lowest similarity index and included the most uncommon genotypes, for which the largest number of specific amplification products were identified. In comparison to the other cultivars, 'Eska 46' was clearly different in terms of its genetic structure, and, overall, few amplification products were generated for it. It should be stressed that in previous studies aimed at evaluating RAPD polymorphism in different populations of *P. pratensis*, such a high genetic diversity among ecotypes had never been found, while the average Dice's coefficient was about 0.7 (*Szenejko & Rogalski, 2015*). Other authors also reported significant genetic relatedness among the different genotypes and cultivars of *P. pratensis* (*Johnson et al., 2002*; *Ning et al., 2005*; *Tamkoc & Arslan, 2010*; *Fard et al., 2012*). UPGMA dendrograms based on genetic similarity matrices revealed discrepancies in the mode of clustering of the studied forms of *P. pratensis*. Greater similarity in that respect was demonstrated for the ISSR and RAPD + ISSR methods. This difference in the clustering process between the RAPD and ISSR dendrograms was also demonstrated in research by *Bhattacharya, Bandopadhyay & Ghosh (2010)*, who were assessing the genetic diversity of *Cymbopogon winterianus* and by *Souframanien & Gopalakrishna (2004)*, who were studying genotypes of *Vigna mungo*. Similarly to our results, these authors found greater clustering conformity for ISSR and RAPD + ISSR. Looking at the distribution in individual UPGMA dendrograms, it appears that the ISSR dendrogram depicts the relationship between the geographical origin of the ecotypes and their genetic similarity in the clearest way. With the exception of populations bie04 189, lbs05 628 and lbs05 170, which differed from the others genetically, the remaining ecotypes were clustered according to their geographical distribution. In the course of PCA, and depending on their geographical origin, a similar tendency in the distribution of the studied populations in two-dimensional space was observed. It was noted that ecotypes from the same voivodships (provinces) or from regions located not far from one another were depicted similarly in two-dimensional space as in the UPGMA dendrogram constructed for the ISSR process. Those populations from eastern Poland and from West

Pomerania and Kujawy clearly differed from the ecotypes originating from the western and southern parts of the country and were located on the opposite side of the horizontal axis of coordinates. They also differed from one another, which can be seen in their arrangement on both sides of the vertical axis. Some discrepancies were found between the UPGMA and PCA clusters of the studied forms, which were related, inter alia, to the distribution of the cultivars. Only a modest relationship between genetic divergence and geographical origin in *P. pratensis* was reported by several authors (*Johnson et al., 2002*; *Fard et al., 2012*; *Raggi et al., 2015*). For comparison, *Abbaszade et al. (2013)*, who were studying the genetic diversity of *Lolium multiflorum*, and *Rahmati et al. (2013)*, who were studying *Festuca arundinacea*, demonstrated a greater result consistency between UPGMA and PCA. However, neither of these research teams was able to demonstrate a correlation between the genetic variability of the grass genotypes they analyzed and their geographical origin.

## CONCLUSIONS

Our results suggest that both the systems of dominant RAPD and ISSR markers that were used in our research can be used to assess the genetic variability of selected ecotypes and cultivars of *P. pratensis*. However, given their better reproducibility of results, greater efficiency in detecting polymorphic amplification products in the presence of a single primer and also their differentiating properties (which make them capable of distinguishing studied forms with respect to their genetic structure), ISSR markers seem to be a more useful and reliable tool for this kind of analysis. The plant material evaluated in this study was characterized by a high genetic variability, which seems to confirm the hypothesis that ecotypes of *P. pratensis* originating from different parts of Central Europe, each with different terrain structures and habitat conditions, can provide great genetic variability and a large collection of alleles. Microsatellites with dinucleotide repeat sequences $(AG)_n$, $(GA)_n$ anchored at the end of 3' proved particularly useful for assessing the genetic diversity of these ecotypes and determining their main directions of variability. PCA helped to establish a correlation between the geographical origin of the studied ecotypes and their genetic variation.

## ACKNOWLEDGEMENTS

We wish to express our gratitude to Anna Kalinka, PhD from the Department of Cell Biology, Faculty of Biology, University of Szczecin for sharing with us her invaluable advice and knowledge regarding the development and interpretation of RAPD and ISSR results.

### Funding

The authors received no funding for this work.

### Competing Interests

The authors declare there are no competing interests.

## Author Contributions

- Magdalena Szenejko conceived and designed the experiments, performed the experiments, analyzed the data, contributed reagents/materials/analysis tools, wrote the paper, prepared figures and/or tables, reviewed drafts of the paper.
- Przemysław Śmietana analyzed the statistical data, prepared figures and/or tables, reviewed drafts of the paper.
- Edyta Stępień analyzed the data.

## Data Availability

The raw data has been supplied as Data S1.

## Supplemental Information

Supplemental information for this article can be found online at http://dx.doi.org/10.7717/peerj.2489#supplemental-information.

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
