# Peer review of "Genetic diversity of Poa pratensis L. depending on geographical origin and compared with genetic markers"

_PeerJ, doi:10.7717/peerj.2489_

## Round 0.1 · original submission · Minor Revisions

Dear authors I consider that your work has been well conducted and designed with valid findings. However, there are some comments raised by the reviewers that need to be taken into account. Your manuscript will benefit from a English Grammar review by an native speaker. Please read and address carefully to all comments made.

Reviewer 1 ·

Basic reporting

No Comments

Experimental design

Experimental design turns out to be discrete and structured. The samples were chosen in a fairly omogenia area of Poland, in the text the coordinates, where the samples have been collected, were reported. The pattern appears to be, therefore, valid even if today to perform cognitive analysis of this type are currently present more reliable markers, innovative simple and easy to use technologies.
The choice of the markers used (RAPD and ISSR) are not tied to the innovative aspect but with little information on the species. It would be interesting to sequence and test the polymorphism found. To be complete, it is recommended to add an image, as the supplemental material, about agarose gels of the most polymorphic primers.

Validity of the findings

The data were well structured, the authors' conclusions indicate a correspondence between geographical origin and genetic diversity for P. pratensis L.. These though in some cases turns out to be true, it should be, however, it associated with different agronomic and biological traits that have lacked in the text or are not considered. We recommend reviewing the findings giving more emphasis to the results demonstrating the validity of its.

Additional comments

Reviewers' comments:

In this manuscript, the authors put their attention on a molecular characterization using RAPD and ISSR on Poa pratensis L.. The authors hypothesize a genetic differentiation of the same depending on their geographical origin, such that if in some ways appears to be true for many species, it should also be relegated to some morphological and biological characteristics of the species. In particular I suggest to the authors, whereas they are in possession of data, if the accessions that were collected and sampled are significant morphological differences. Certainly, the purpose of this study is important and it complies with the scope of the "PeerJ".


Point-by-point comments

Keywords, line 34
Some keywords should be replaced in order to be different from the title.

Materials and Methods, line 90:

Modify the sentence as follows; I'd suggest the authors to check more carefully the punctuation of phrases and sentences.

>>The assessment of genetic diversity was performed for three cultivars used for different purposes, including 'Eska 46', one of the oldest Polish fodder cultivar and two lawn cultivars: ‘Limousine’, an apomictic uniclonal German cultivar, and 'Alicia', the oldest Polish lawn cultivar.<<

Materials and Methods, line 101: >> 69 primers were tested […] <<

I would recommend that authors take great care not to begin sentences with a number, in case of writing it in words, i.e. >> Sixty-nine primers […] <<

Materials and Methods, line 103: >> T100TM Thermal Cycler (Bio-rad ……). <<

Insert after "Bio-rad" information related to the company which city and / or country as the authors have done previously.

Materials and Methods, line 105-106: >> The reactions were carried out on a total volume
of 25 μl containing the following ingredients: […] <<

Materials and Methods, line 118-120: >> The resulting RAPD and ISSR amplification products were separated on 2.0% agarose gel with ethidium bromide (5μg/ml; Sigma-Aldrich) in TAE buffer for 4-5 hours. >>
I' suggest you move down the paragraph in a new chapter or in “Data analysis”.

Results, line 174: The word "obtained" is repeated too many times. I suggest you look for the appropriate synonym and make the sentence longer-side sliding to reading.


There are so many lacks of space throughout the manuscript and many words are connected as I pointed out here. The authors should have checked the manuscript before submission.


Results, line 234-235: Modify the sentence as follows; I'd suggest the authors to check more carefully the punctuation of phrases and sentences. >> separately for RAPD, ISSR and (RAPD+ISSR). <<

Results, line 253 >> ISRR << the author wrote ISRR instead of ISSR, I suggest you check the text to see if there are other errors of this type.


Discussion, line 288 >> proved to be most […]<<

The authors meant to say "the most" or "more"?

·

Basic reporting

There are minor grammatical problems throughout the text that will require correction, presumably by the technical editor. The most common issue is a need for additional punctuation marks, primarily commas, to divide sentences up into unambiguous clauses. There were also multiple instances of sentences beginning with numbers not written out as words. Otherwise the "basic reporting" standards were met.

Experimental design

The experimental design standards have been met.

Validity of the findings

The "validity of the findings" standards have been met.

Additional comments

One approach to improving the grammar and sentence structure would be to engage in several rounds of translation from Polish to English to Polish to English to see how well constructed your sentences really are. Those that fail to survive in easily comprehended formats would be prime targets for rewriting. The overall length is slightly too long for the amount of data that you present. Be judicious in chosen which points are worth elaborating on in greater detail and which can and should be said more succinctly.

---

## Round 0.2 · accepted · Accept

Dear authors

Thanks for taking into consideration all comments raised by the reviewers. I think that the manuscript now reads nicely.